# Carrier-Free Inhalable Dry Microparticles of Celecoxib: Use of the Electrospraying Technique

**DOI:** 10.3390/biomedicines11061747

**Published:** 2023-06-17

**Authors:** Azin Jahangiri, Ali Nokhodchi, Kofi Asare-Addo, Erfan Salehzadeh, Shahram Emami, Shadi Yaqoubi, Hamed Hamishehkar

**Affiliations:** 1Department of Pharmaceutics, School of Pharmacy, Urmia University of Medical Sciences, Urmia 571579-9313, Iran; emami.sh@umsu.ac.ir; 2Pharmaceutics Research Laboratory, School of Life Sciences, University of Sussex, Brighton BN1 9QJ, UK; 3Lupin Inhalation Research Center, Lupin Pharmaceuticals Inc., Coral Spring, FL 33065, USA; 4Department of Pharmacy, University of Huddersfield, Huddersfield HD1 3DH, UK; k.asare-addo@hud.ac.uk; 5Student Research Committee, School of Pharmacy, Urmia University of Medical Sciences, Urmia 571579-9313, Iran; sa.erfan@gmail.com; 6Biotechnology Research Center, and Research Center for Integrative Medicine in Ageing, Tabriz University of Medical Sciences, Tabriz 516661-5731, Iran; yaqoubish@tbzmed.ac.ir; 7Drug Applied Research Center, Tabriz University of Medical Sciences, Tabriz 516661-6471, Iran; hamishehkar.hamed@gmail.com

**Keywords:** dry powder inhaler, Celecoxib, electrospraying, aerosol performance, carrier-free DPI

## Abstract

Upregulation of cyclooxygenase (COX-2) plays an important role in lung cancer pathogenesis. Celecoxib (CLX), a selective COX-2 inhibitor, may have beneficial effects in COVID-19-induced inflammatory storms. The current study aimed to develop carrier-free inhalable CLX microparticles by electrospraying as a dry powder formulation for inhalation (DPI). CLX microparticles were prepared through an electrospraying method using a suitable solvent mixture at two different drug concentrations. The obtained powders were characterized in terms of their morphology, solid state, dissolution behavior, and aerosolization performance. Electrosprayed particles obtained from the ethanol–acetone solvent mixture with a drug concentration of 3 % *w*/*v* exhibited the best in vitro aerosolization properties. The value of the fine particle fraction obtained for the engineered drug particles was 12-fold higher than that of the untreated CLX. When the concentration of CLX was increased, a remarkable reduction in FPF was obtained. The smallest median mass aerodynamic diameter was obtained from the electrosprayed CLX at a 3% concentration (2.82 µm) compared to 5% (3.25 µm) and untreated CLX (4.18 µm). DSC and FTIR experiments showed no change in drug crystallinity or structure of the prepared powders during the electrospraying process. The findings of this study suggest that electrospraying has potential applications in the preparation of DPI formulations.

## 1. Introduction

Increased expression of cyclooxygenase-2 (COX-2) is common in non-small cell lung cancer (NSCLC) and is associated with a poor prognosis. Celecoxib (CLX), a specific COX-2 inhibitor, induces apoptosis in human NSCLC cells via the extrinsic death receptor pathway [1]. This drug also has beneficial effects on lung cancer chemoprevention in former smokers [2]. Additionally, CLX may also be considered a valuable adjunct in the treatment of coronavirus disease (COVID-19)-triggered pulmonary and systemic inflammatory storms [3,4,5,6]. Considering the above facts, the direct delivery of CLX to the lung is beneficial and can provide higher concentrations in the lungs with lower systemic side effects [7].

Pulmonary drug delivery as a well-established approach is gaining increasing attention for both the local pulmonary and systemic delivery of drugs [8]. There are three main approaches for a drug to become delivered to the lungs, namely, nebulizers, metered dose inhalers (MDI), and dry powder inhalers (DPIs) [9]. The lower amount of excipients along with the higher physical and chemical stability of DPI formulations makes them more prominent candidates for pulmonary drug delivery, especially for high-dose drugs [10]. Furthermore, in comparison to the liquid formulations, DPIs have become increasingly attractive for lung delivery because of improved portability, stability, and potentially superior patient adherence [11]. Undoubtedly, dry powders in the micron range are considered important components of drug delivery systems, as they are used to deliver drugs to specific sites in the body, improve drug stability and bioavailability, and provide a controlled release of drugs over a period of time. Dry powders can be administered via inhalation, oral, or injectable routes, depending on the drug and the desired therapeutic effect [12,13,14,15].

Regarding the rapidly growing popularity of pulmonary delivery, the production of tailor-made inhalable drug particles with higher delivery efficiency to the lungs and improved therapeutic efficiency seems crucial. To achieve ideal formulations for pulmonary delivery, a wide variety of novel particle engineering techniques have been developed over the past decade [8].

DPI formulations are prepared using particle engineering procedures, such as electrospraying [16], spray drying [12], milling [17] supercritical fluid [18], and crystallization [19]. Electrospraying or electrohydrodynamic (EHD) spraying is a technique that generates fine droplets of charged particles from a solution or suspension by applying an electric field. In this process, a solution containing API/s and/or excipients is atomized by electrical forces. Thus, the liquid is exposed to electrical shear stress by maintaining the nozzle at a high electric potential [20]. The production of extremely small droplets with controlled size, charge, porosity, and morphology makes electrospraying an emerging technology for producing micro/nano-sized particles, especially thermo-sensitive compounds, such as bioactive compounds [21,22,23,24,25]. Although electrospraying is rarely used for the manufacturing of inhalable particles, this technique has great potential as a particle engineering procedure in the production of inhalable lactose with significantly increased fine particle fraction (FPF) [16]. The applied ambient temperature in electrospraying allows the usage of a wide variety of organic solvents to modulate dry powders with preferred characteristics towards better performance.

The current state-of-the-art in electrospraying for pulmonary delivery involves the use of optimized particles that could effectively reach the target area in the lungs. Central airways can be targeted with 3–5 μm aerosols to treat local lung diseases, such as acute bronchoconstriction, whereas smaller particles (1–3 μm) are used for deep lung deposition and subsequent systemic absorption [26,27,28]. Researchers are working on developing new formulations of drugs and other therapeutic compounds that can be effectively administered using this method [29]. Research is underway to develop new materials for electrospraying, such as biodegradable polymers, to reduce potential toxicity and improve therapeutic efficacy [30,31]. Some studies have also explored the use of electrospray in combination with other techniques, such as microfluidics, to further optimize the particle size and distribution [32]. The fields of electrospraying and pulmonary delivery are constantly evolving. Ongoing research is aimed at developing new and improved approaches to effectively deliver therapeutic compounds into the pulmonary system.

Considering the advantages of the electrospraying technique for particle engineering and the usefulness of pulmonary drug delivery of CLX, for the first time, the aim of the present study was to utilize this technique for the preparation and tailoring of carrier-free inhalable microparticles of CLX suitable for DPI formulations. The physicochemical, dissolution, and aerosol properties of the generated microparticles were evaluated to ascertain whether they had any superior properties.

## 2. Materials and Methods

Celecoxib (CLX) (PubChem CID: 2662) was purchased from Tehranchemie Company (Tehran Chemistry Co., Tehran, Iran). Ethanol, acetone and sodium dodecyl sulphate (SDS) were purchased from Merck (Merck Co., Shanghai, China). Water was used in double-distilled quality from the lab. All other materials were of analytical grade.

### 2.1. Preparation of DPI Powders by Electrospraying

The electrospraying experimental setup contained a syringe pump (Fanavaran Nano-Meghyas Co., Mashhad, Iran), a stainless steel 23 G blunted needle (0.6 mm inner diameter), a high voltage generator (Fanavaran Nano-Meghyas Co., Mashhad, Iran), and a stainless-steel collector plate. The solvent system and feed concentrations were selected, based on our previous work on electrospraying of other drugs [10,33]. The selected solvent should be volatile, have sufficient electrical conductivity, and show good solvency for CLX. Based on these criteria, an ethanol-acetone (1:1) mixture was used to dissolve CLX. CLX was completely dissolved in the solvent system at room temperature to achieve 3% and 5% *w*/*v* concentrations, and it was stirred for 15 min at 200 rpm on a hot-plate stirrer (Heidolph MR Hei-Tec magnetic stirrer, Heidolph Co., Schwabach, Germany) at room temperature. Electrospraying of the prepared solution was conducted under the following conditions: feeding rate 1.5 mL/h, working voltage 20 kV, and tip-to-collector distance of 17 cm. These experimental conditions used were initially optimized by varying factors, such as the drug solution concentration under several electrospraying conditions. The prepared particles were removed from the collector plate and kept in a desiccator (to protect them from humidity) at room temperature (20–25 °C) until further analyses (Figure 1).

### 2.2. Scanning Electron Microscopy (SEM)

The shape, surface morphology, and size distribution of the particles were studied by a field emission scanning electron microscope instrument (TESCAN MIRA, Brno–Kohoutovice Czech Republic) at accelerating voltages of 10–20 kV. Samples were sputter-coated with gold (DST1 model, Nanostructured Coating Co., Tehran, Iran) prior to examination.

### 2.3. Fourier-Transform Infrared Spectroscopy (FTIR)

Fourier-transform infrared spectroscopy (FTIR) was used to investigate possible changes in the chemical structure of the drug during the electrospraying process using the KBr disc method. A PerkinElmer spectrometer (PerkinElmer Co., Shelton, CT, USA) was used to record IR spectra over a spectral region from 4000 to 400 cm^−1^ with a resolution of 2 cm^−1^.

### 2.4. Differential Scanning Calorimetry (DSC)

Thermal analysis was performed using DSC (S800-SPICO, Tehran, Iran) to investigate possible changes in the thermal behavior of the drug during the electrospraying process. Approximately 5 mg samples were placed into aluminium pans and sealed hermetically. The samples were scanned from 25 to 300 °C at a heating rate of 10 °C/min under nitrogen atmosphere.

### 2.5. In Vitro Aerosolization Assessment

A Next Generation Impactor (NGI) (COPLEY scientific, United Kingdom) equipped with an induction port and a pre-separator was utilized to study the in vitro aerosolization performance of the prepared powder formulations. Before the emission, 15 mL of solvent was incorporated into the pre-separator, and the seven collection cups and micro-orifice collector (MOC) were coated with a solution of Tween 80 in ethanol (1% *w*/*v*). DPI samples were prepared by filling an average amount of 10 mg of untreated CLX as the control and electrosprayed CLX powders into size 3 capsules (Pure Capsules, DR T&T Health UK Ltd., Corby, UK). To aerosolize the formulation from the capsule, the prepared capsule was placed in the DPI device (Aerolizer^®^, Novartis, Basel, Switzerland) and was connected to the mouthpiece adaptor of the NGI. A high-capacity pump (HCP5, Copley Scientific, Nottingham, UK) was attached to TPK (Copley, UK) to control the flow rate at 100 L/min. The flow rate was measured using a flow meter (DFM 2000, COPLEY Scientific, UK). Before the deposition test, the capsules were pierced using the pins located inside the DPI device. The number of actuations for each formulation was 1 with a 2.4 s duration. The drug remained in the inhaler device and capsule, and the drug deposited in the USP induction port, pre-separator, seven collection cups, and MOC were rinsed with 4, 10, 10, and 4 mL of solvent (ethanol), and they were determined spectrophotometrically using a Cecil UV/VIS spectrophotometer (λmax = 253 nm). The resulting data were analyzed using Copley inhaler testing data analysis software (CITDAS), (Version 3.10 Wibu (USP 32/Ph. Eur. 6.0) COPLEY Scientific, Nottingham, UK).

### 2.6. In Vitro Dissolution Studies

In addition to the deposition patterns, the dissolution of drug particles in the lung is necessary for membrane absorption [34]. A dissolution rate study was performed on untreated CLX and electrosprayed CLX with 3% concentration (the optimized formulation with better aerosolization performance) using USP dissolution apparatus 2 (Pharmatest Co, Hainburg, Germany). Samples equivalent to 20 mg CLX were placed in the dissolution medium (500 mL of 0.5% *w*/*v* SLS containing distilled water which was maintained at 37 ± 0.5 °C under a stirring rate of 50 rpm). Dissolution tests were performed in triplicate at predetermined time intervals (3, 6, 10, 15, 30, and 60 min) for each formulation. The concentrations of the dissolved drug were measured spectrophotometrically (Cecil, UK) at 253 nm, and then the percentages of dissolved CLX were plotted against time.

## 3. Results

### 3.1. Preparation of Dry Powder Formulations by Electrospraying

After preliminary studies, CLX microparticles were successfully prepared using the ethanol-acetone (1:1) solvent mixture with the electrospraying method at room temperature (20–25 °C), where the concentration of CLX was adjusted to 3% and 5% *w*/*v* concentrations. The electrospraying parameters comprising nozzle diameter, electrical voltage, flow rate, and working distance were optimized to form a stable and continuous cone-jet mode [35]. In general, within the normal limits, lower nozzle diameter, higher electrical potential, higher flow rate, and lower distance between the collector and the nozzle tip can lead to the creation of smaller droplets and particles, whereas higher concentrations of drug solution can result in the production of larger particles [36,37,38]. It is noteworthy to mention that, for drug solutions with high surface tension values, a higher electrical voltage is required to defeat the interface tension forces and to break down the liquid into small droplets in the jet [38].

### 3.2. Particle Size and Morphology

Aerosol performance is highly associated with the size and shape of the prepared particles. Accordingly, these properties should be carefully modulated when designing DPI formulations [39]. SEM micrographs of the untreated CLX and the prepared particles (electrosprayed solutions containing 3% and 5% *w*/*v* CLX) are shown in Figure 2. In general, the size of the particles formed is greatly dependent on the concentration of the electrosprayed solution [38]. For instance, when the concentration of CLX in electrosprayed solutions increased from 3 to 5%, the particle size became larger. The SEM micrographs presented that the size and morphology of the CLX powder were changed during the electrospraying process (Figure 2). Unlike untreated CLX powder, which exhibited needle-like crystals with lengths ranging from 25 to 527 µm (Figure 2A), the electrosprayed samples displayed a smaller size range (about 3 µm) with different morphological appearance. The SEM images revealed that the concentration of CLX used during the electrospraying process has an impact on the morphology of the obtained particles. For instance, when the concentration of CLX was 3%, most of the obtained particles were rod-like in shape (Figure 2B), whereas, in the case of the 5% CLX, the produced particles had irregular plate-like morphology (Figure 2C).

### 3.3. Solid State Characterizations

The thermal behavior of untreated CLX, as well as CLX particles obtained from the electrospraying of 3% and 5% CLX solutions, are illustrated in Figure 3. The DSC traces of untreated CLX particles obtained from electrosprayed solutions of CLX (3% and 5 % *w*/*v*) showed an endothermic peak at 165.3, 162.9, and 164.7 °C, respectively. As can be seen, the melting points of the electrosprayed samples were similar to those of untreated CLX (as in previous studies) [40].

FTIR spectroscopy was applied for the assessment of potential structural changes in CLX following the electrospraying process. The FTIR spectrum of untreated CLX was compared with that of the electrosprayed CLX samples (Figure 4). The observed characteristic peaks of untreated CLX at 1163 and 1350 cm^−1^ were attributed to S=O symmetric and asymmetric stretching, respectively. Medium intensity bands at 3338 and 3242 cm^−1^ were seen as a doublet, which is assigned to the N–H stretching vibration of the –SO_2_ NH_2_ groups [41,42]. As shown in Figure 4, no visible shift in peaks was observed in the FTIR spectra of the prepared samples compared to that of the untreated CLX. This indicates that there was no change in the chemical and crystal structure of the prepared drug formulations after preparation through the electrospraying process.

### 3.4. In Vitro Deposition Studies

The in vitro aerosolization efficiency of the CLX DPI formulations was evaluated by measuring inhalation parameters, such as mass median aerodynamic diameter (MMAD), geometric standard deviation (GSD), fine particle dose (FPD) and fine particle fraction (FPF). The amount of drug in percentage deposited in each stage of the NGI (aerosolization behavior) for the various formulations is given in Figure 5. The data in Figure 5 were used to calculate aerosolization efficiency parameters in Table 1. As shown in Table 1, the FPF% of both electrosprayed formulations was significantly higher (49.83% and 24.42% for 3 and 5% CLX solutions, respectively) than that of the untreated CLX (4.17%). The MMAD results indicated that all treated and untreated CLX particles were in the inhalable size range (<5 μm). However, the electrosprayed CLX formulation obtained from the 3% CLX solution exhibited the smallest MMAD with the narrowest size distribution (based on the GSD values in Table 1). On the other hand, although the MMAD value for the untreated CLX was acceptable (4.73 µm), it was significantly higher than the MMAD values of the electrosprayed formulations, hence demonstrating a relatively poorer performance compared to the electrosprayed formulations. A comparison of the in vitro inhalation profiles of the two electrosprayed formulations showed that the formulation containing engineered CLX obtained from the 3% drug solution had better lung drug delivery performance than the CLX formulation obtained from the electrospraying of the 5% *w*/*v* drug solution. This is evident from the amount of drug deposited in stages 3, 4, and 5 of the NGI (Figure 5).

### 3.5. In Vitro Dissolution Studies

The importance of the dissolution of orally inhaled drug formulations in DPIs was highlighted by Nokhodchi et al. [43]. Pulmonary drug absorption and, consequently, clinical performance, might be limited by the dissolution or release processes. Therefore, an in vitro dissolution rate (especially for poorly water-soluble drugs) is an important parameter that can influence in vivo performance [44]. Release profiles of electrosprayed CLX particles (3% *w*/*v* was selected due to showing the best performance compared to the 5% *w*/*v* CLX solution), as well as untreated CLX, are demonstrated in Figure 6. As can be observed, both samples showed a similar release profile. Despite the smaller size of the electrosprayed CLX, it did not show an improved dissolution profile compared to untreated CLX. This could be attributed to the poor wetting and aggregation of the hydrophobic submicron particles during the dissolution process [45].

## 4. Discussion

Considering the importance of CLX pulmonary delivery as a targeted delivery method with a more rapid onset, only limited research has been conducted on the inhalable forms of CLX. Patolla et al. showed that the aerosolization performance of CLX-encapsulated nanostructured lipid carriers improved CLX pulmonary bioavailability compared with the solution formulation, which could potentially lead to better patient compliance with minimal dosing intervals [46]. In another study, it was shown that large porous celecoxib–PLGA microparticles prepared using supercritical fluid technology showed sustained drug delivery and anti-tumor efficacy without producing any significant toxicity [47]. Simultaneous pulmonary administration of celecoxib and naringin through a nebulizing nanoemulsion showed controlled in vitro release profiles and good aerosolization performance, with superior cytotoxicity in A549 lung cancer cells, which can be useful in the treatment of pulmonary diseases [48].

Owing to the high therapeutic dose of CLX, the development of carrier-free inhalable particles is preferable to carrier-containing DPIs. Designing DPI formulations without any carrier (which is the major component of DPI formulations, usually above 99% of the DPI composition) makes them preferable in terms of a lower amount of inactive substances, and subsequently fewer adverse effects, lower production costs, and higher drug loading capacity [49]. Notably, the pulmonary deposition of carrier-free particles is more likely due to better powder homogeneity and aerosolization, in addition to non-intraparticle forces between the carrier and drug. Furthermore, it is easy to combine two or three active ingredients in the same powder [50].

Carrier-free particles are mainly produced by spray drying [50]. Other techniques, such as spray freeze-drying, supercritical fluids, air jet micronization, and atomic layer deposition, are also available and are already being explored for pulmonary delivery by others [51,52,53,54,55]. Despite the popularity of spray drying as an effective method for drying a wide range of products and pharmaceuticals, it can result in product damage and is costly to build and operate [56,57,58].

Electrospraying is an emerging multipurpose device for the drying of solutions and suspensions using electrical forces. This technique is simple, economical, has a high production rate, is applicable at room temperature, and is promising for the drying and encapsulation of sensitive compounds. This technique can be performed by adjusting the process variables, such as solution properties, solvent type, and process parameters (applied voltage as well as distance between the needle tip and collector plate, applied flow rate, and ambient temperature and humidity). This technique is widely used in particle engineering for the production of polymer-based and polymer-free micro-and nanoparticulate drug delivery systems with desired properties [31,59]. In recent years, this method has also been effectively used for pulmonary drug delivery [10,28,60].

In this study, we aimed to prepare carrier-free inhalable CLX microparticles as DPI formulations using an electrospraying method. The desired CLX microparticles were successfully prepared by a one-step electrospraying process using an ethanol-acetone (1:1) solvent mixture at two different concentrations of the CLX solution (3% and 5% *w*/*v*).

SEM micrographs revealed that the concentration of CLX in the electrosprayed solutions affected the particle size and morphological appearance of the prepared powder; thus, an increase in CLX concentration from 3 to 5% led to the production of larger particles with different morphologies. Moreover, electrosprayed particles at 3% concentration showed rod-like morphology, whereas 5% concentration led to the production of an irregular plate-like appearance. This indicates that, by changing the concentration of CLX, the morphology and size of the particles can be modulated, thereby affecting the aerosolization performance and pharmaceutical processability [61,62,63]. Previous studies demonstrated that rod-shaped particles with a high aspect ratio showed improved aerodynamic properties, as well as less macrophage uptake compared to spherical-shaped particles. Moreover, preferential uptake of rod-shaped particles by small cell lung cancer cells was also observed. It is noteworthy that in vitro tumor simulation studies have shown that rod-shaped particles display boosted anti-tumorigenic activity with less cardiotoxicity [64].

Solid-state characterization of the electrosprayed samples revealed that the melting points of the electrosprayed CLX at both 3% and 5% concentrations were almost similar to those of the untreated samples. This similarity indicates that the crystalline structure of the drug remained stable and unchanged during the electrospraying process. The shift in the melting point of any drug to lower temperatures could be attributed to the change in the crystalline structure during the electrospray process, which was not the case for CLX particles in the current study. In the electrospraying process, the solvent rapidly evaporates from the drug solution, and the crystallized particles are deposited on the collector screen. This rapid solvent removal can sometimes lead to a change in crystallinity. Several studies have reported that electrospraying can alter the crystallinity of polymers (poly(L-lactic acid), polytetrafluoroethylene) and drugs (budesonide) [10,65,66]. According to the FTIR results, no structural changes were observed between the electrosprayed samples and the untreated CLX. This confirmed that the chemical and crystal structures of the drug remained stable and did not undergo any changes during the electrospraying process.

An in vitro aerosolization study showed that the FPF% of both electrosprayed formulations was significantly higher (49.83 and 24.42% for 3 and 5% CLX solutions, respectively) than that of untreated CLX (4.17%). The poor in vitro aerosolization performance (FPF = 4.18%) of the untreated formulation could be related to the very large geometric diameter of the untreated CLX particles, leading to the inertial impaction of these particles in the upper airways [67]. The electrosprayed CLX formulation at 3% concentration showed the highest FPF%, and it showed the smallest MMAD with the narrowest size distribution (according to the GSD values in Table 1). These observations could be correlated to the elongated morphology of electrosprayed CLX at a concentration of 3%. This is in agreement with published data reporting the more acceptable inhalation performance of particles with an enhanced elongation ratio [68], which is attributed to the enhanced floating time of the particles in the respiratory tract and the deposition of these particles in the lower airways. This was also the case when elongated carriers, such as lactose and mannitol, were used in DPI formulations containing salbutamol sulfate [69]. Comparing the FPF, MMAD, FPD, and GSD values of the electrosprayed formulations, the formulation containing CLX obtained via the 3% *w*/*v* CLX solution showed the best performance.

No significant improvement was observed in the dissolution profile of the electrosprayed particles obtained using the 3% *w*/*v* CLX solution compared to that of the untreated CLX, despite their relatively smaller particle size. This may be related to the poor wetting of hydrophobic submicron particles, which could result in aggregation during the dissolution process [45]. This aggregation tendency may affect the dissolution of these particles. Although fast dissolution could be an advantage for orally inhaled drugs, where systemic absorption is required, in the case of locally acting drugs, the aim is to retain the drug particles longer in the lungs for better in vivo performance.

## 5. Conclusions

In conclusion, our results revealed the feasibility of using electrospraying as a novel platform for the production of inhalable microparticles of CLX in a single step without the need for additives. The aerodynamic properties of the produced particles were significantly affected by feed concentration, where the electrosprayed CLX at a 3% concentration showed the smallest mass median aerodynamic diameter (2.82 µm) compared to untreated CLX (4.18 µm). This was reflected in fine particle fraction values where formulations with the smallest MMAD value showed the highest FPF of 49.83%, and untreated CLX showed the lowest FPF of 4.18%. The results also showed that, when the concentration of CLX in electrosprayed formulation increased from 3% to 5%, the value of FPF decreased from 49.83% to 24.43% (MMAD of this formulation was 3.25 µm). From a stability perspective, electrospraying did not compromise the chemical and crystalline structures of CLX. An unexpected result was the low dissolution rate of micronized particles. To address this challenge, further studies should focus on resolving the agglomeration tendency of the electrosprayed particles.

## Figures and Tables

**Figure 1 biomedicines-11-01747-f001:**
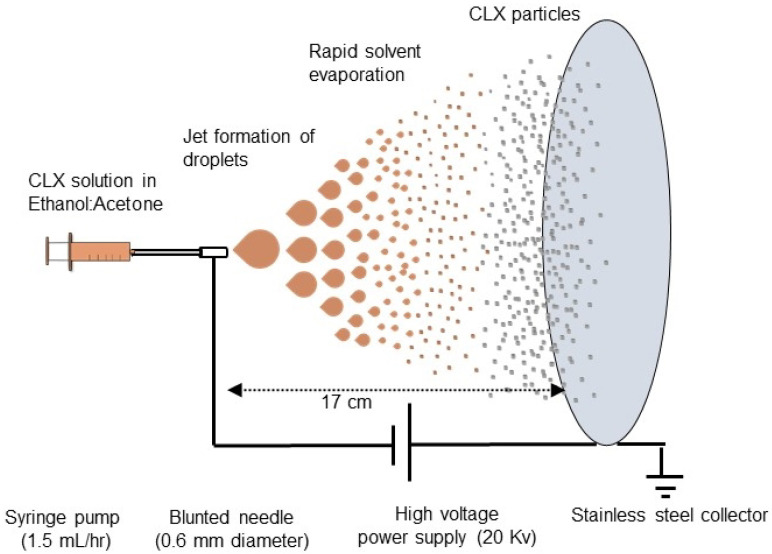
Schematic representation of the electrospraying process.

**Figure 2 biomedicines-11-01747-f002:**
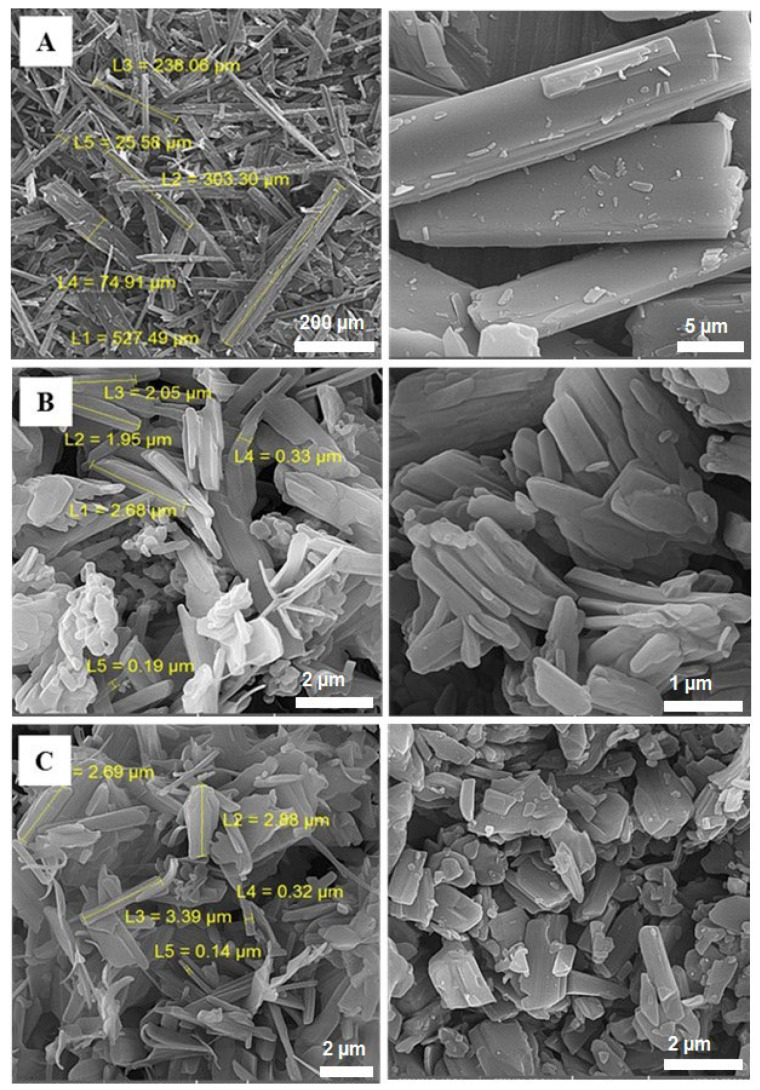
SEM micrographs of (**A**) untreated CLX, (**B**) particles obtained from electrosprayed solution containing 3% CLX, and (**C**) 5% CLX at different magnifications.

**Figure 3 biomedicines-11-01747-f003:**
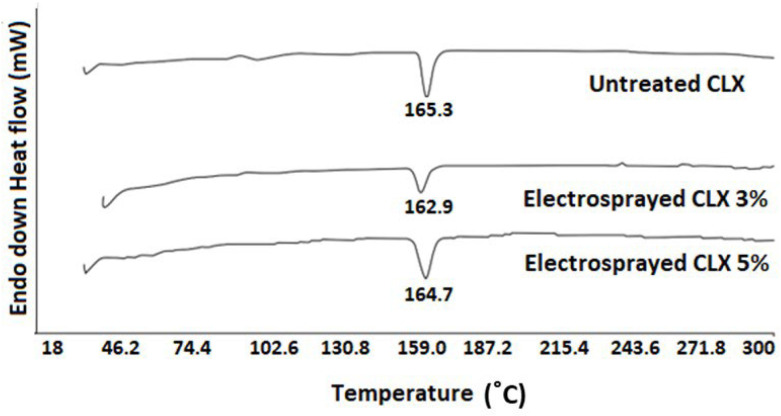
DSC thermograms of untreated CLX, as well as electrosprayed CLX particles obtained from 3% and 5% *w*/*v* CLX solutions.

**Figure 4 biomedicines-11-01747-f004:**
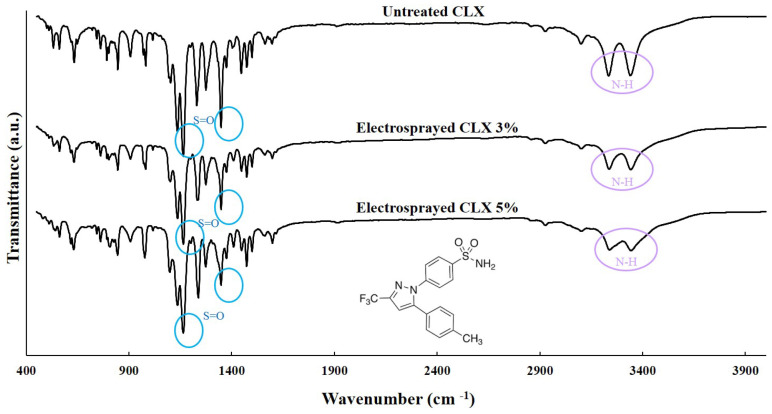
FTIR spectra of untreated CLX, as well as electrosprayed CLX obtained from solutions containing 3% and 5% *w*/*v* CLX. The chemical formula presented in the figure shows the chemical structure of CLX.

**Figure 5 biomedicines-11-01747-f005:**
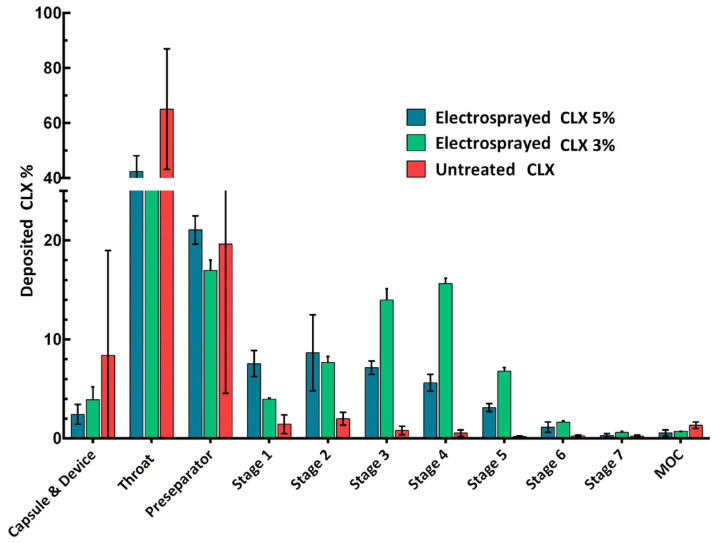
In vitro deposition of untreated CLX, as well as electrosprayed CLX particles obtained from 3 and 5% *w*/*v* CLX solutions.

**Figure 6 biomedicines-11-01747-f006:**
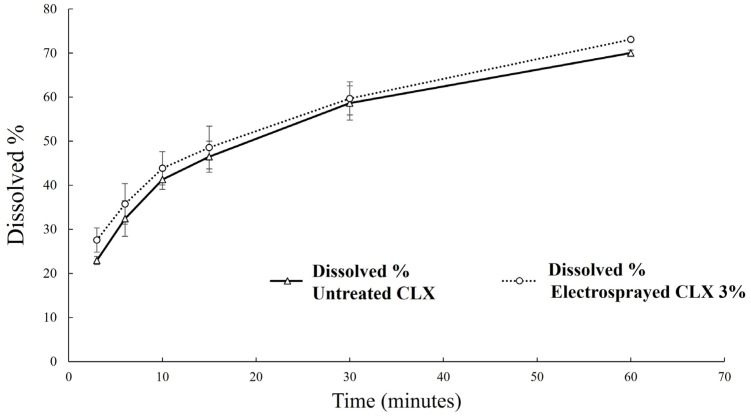
The dissolution profiles of untreated CLX and electrosprayed CLX in 3% *w*/*v* concentration.

**Table 1 biomedicines-11-01747-t001:** Mass Median Aerodynamic Diameter (MMAD), Geometric Standard Deviation (GSD), Fine Particle Dose (FPD), and Fine Particle Fraction (FPF) of the untreated CLX and electrosprayed samples (mean ± SD, *n* = 3).

In Vitro Aerosolization Parameters	Untreated CLX	Electrosprayed CLX (3 *w*/*v*)	Electrosprayed CLX (5 *w*/*v*)
FPD (µg)	236.21 ± 21.6	1323.76 ± 98.2	762.57 ± 53.1
FPF (%)	4.18 ± 1.3	49.83 ± 12.1	24.43 ± 6.2
MMAD (µm)	4.73 ± 1.3	2.83 ± 0.09	3.25 ± 0.2
GSD	NA	1.97 ± 0.05	2.41 ± 0.01

## Data Availability

Additional available data can be obtained from the authors upon request.

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
