# Peer review of "Carrier-Free Inhalable Dry Microparticles of Celecoxib: Use of the Electrospraying Technique"

_biomedicines, 2023, doi:10.3390/biomedicines11061747_

Round 1
Reviewer 1 Report
This manuscript provide preparation methods for carrier-free inhalable microparticles of celecoxib. Although the work is not fully comprehensive. the presented data have some contributions to the related research fields. From the latter positive viewpoint, I may suggest publication of this work in Biomedicines. However, some revisions are necessary. Please see below.
1) It is better to add the initial figure to explain the used materials and their fabirication methods.
2) Addition of one final figure to visually represent discussion part would be helpful for further understanding.
3) The current title may give common and less innovative impression. Including a new concept word in the title would increase innovative impression of the work itself. I may suggest use of an emerging conceptual term, nanoarchitectonics, in the title (as post-nanotechnology concept, see https://pubs.rsc.org/en/content/articlelanding/2021/nh/d0nh00680g). For example, the title like ... Electrospray nanoarchitectonics of carrier-free inhalable microparticles of celecoxib ... may sound more innovative.
4) In Figure 3, Transmittance (%) of the vertical parameter should be Transmittance (a.u.).
5) Too detailed error values are not meaningful. Usually error values are represented in one or two order.
Author Response
Refer to the attached file.

Reviewer 2 Report
The authors should considerably improve the "Introduction" section in order to give to reader the state of the art in this area. This section is an opportunity to provide readers with the background necessary to understand the paper. A well-written, documented introduction will broaden your readership by making your findings accessible to a larger audience.
Please sharpen the description of the novelty factor in the "in this work" section of the introduction. What exactly was done in this study for the first time? It is important to put this article in perspective to enable readers to quickly decide whether to read the paper or not. How did you choose the 3% and 5%ww solution?
Materials and methods section is too long and should be more compacted, while keeping the most important information. This section is written too general as a kind of report, and not suitable for a scientific paper. Revise the synthesis section in a more “technical” way. Please standardize the details regarding all used equipment (such as name, model, manufacturer, measurement conditions, etc.) for measurements.
Results. Figure 1. The scale bar is not visible. Figure 3. The chemical formula of the CLX should be presented in order to highlight the mentioned groups. Figure 4. Please use abbreviation CLX.
The discussion presented is poor, in terms of discussing its results and comparing them with the bibliography. I suggest reviewing this part more carefully and discuss. The existing literature data refer mainly to general aspects.
Conclusions section is an important part of the paper; it delivers closure for the reader while reminding the reader of the contents and importance of the paper. It accomplishes this by stepping back from the specifics in order to view the bigger picture of the document.
Finally, I believe that the work is not suitable for publication in this form and requires large revision.
Minor corrections.
Author Response
Refer to the attached file.

Round 2
Reviewer 1 Report
Replies and revisions are fine. The revised version becomes acceptable.
Author Response
Thanks for accepting the revised manuscript.
Reviewer 2 Report
Based on this revised version of the manuscript, there are still unanswered questions and weaknesses in this paper.
1. The authors should considerably improve the "Introduction" section in order to give to reader the state of the art in this area. This section is an opportunity to provide readers with the background necessary to understand the paper. A well-written, documented introduction will broaden your readership by making your findings accessible to a larger audience.
2. The conclusion section should have the main results in quantitative statements as well. Please improve.
Author Response
Refer to the attached file.

Round 3
Reviewer 2 Report
The authors responded to reviewer suggestions and improved the manuscript therefore I recommend printing it in its present form.
Minor editing of English language required.